# ARE GRAPH CONVOLUTIONAL NETWORKS FULLY EXPLOITING THE GRAPH STRUCTURE?

## ABSTRACT

Graph Convolutional Networks (GCNs) represent the state-of-the-art for many graph related tasks. At every layer, GCNs rely on the graph structure to define an aggregation strategy where each node updates its representation by combining information from its neighbours. A known limitation of GCNs is their inability to infer long-range dependencies. In fact, as the number of layers increases, information gets *smoothed* and node embeddings become indistinguishable, negatively affecting performance. In this paper we formalize four levels of injection of graph structural information, and use them to analyze the importance of long-range dependencies. We then propose a novel regularization technique based on random walks with restart, called RWRReg, which encourages the network to encode long-range information into node embeddings. RWRReg does not require additional operations at inference time, is model-agnostic, and is further supported by our theoretical analysis connecting it to the Weisfeiler-Leman algorithm. Our experimental analysis, on both transductive and inductive tasks, shows that the lack of long-range structural information greatly affects the performance of state-of-the-art models, and that the long-range information exploited by RWRReg leads to an average accuracy improvement of more than $5\%$ on all considered tasks.

## 1 INTRODUCTION

Graphs are a ubiquitous data representation of many real world phenomena, with applications ranging from social networks, to chemistry, biology, and recommendation systems (Zhou et al., 2018). Graph Neural Networks (GNNs) are the generalization of deep learning for graph structured data, and have received a huge amount of attention from the research community.

One class of GNN models, the Graph Convolutional Network (GCN), has demonstrated to be extremely effective and is the current state-of-the-art for tasks such as graph classification, node classification, and link prediction. GCNs adopt a message passing mechanism where at each layer every node in the graph receives a message (e.g. a feature vector) from its 1-hop neighbours. The massages are then aggregated with a permutation invariant function (e.g. by mean or sum) and are used to update the node's representation vector with a learnable, possibly non-linear, transformation. The final node embedding vectors are used to make predictions, and the whole process is trained end-to-end. Empirically, the best results are obtained when the message passing procedure is repeated 2 or 3 times, as a higher number of layers leads to over-smoothing (Li et al., 2018; Xu et al., 2018b). Thus, GCNs are only leveraging the graph structure in the form of the 2-hop or 3-hop neighbourhood of each node. A direct consequence of this phenomenon is that GCNs are not capable of extracting and exploiting long-range dependencies between nodes.

Random walks with restart (Page et al., 1998) have proven to be very effective at quantifying how closely related two nodes are (Tong et al., 2006), regardless of their distance in the graph. In fact random walks with restart can capture the global structure of a graph, and have been used for many tasks including ranking, link prediction, and community detection (Jin et al., 2019). On the other hand, random walks with restart do not consider node features, which are instead heavily exploited by GCNs. Combining GCNs and random walks with restart could then provide a powerful method to fully exploit the information contained in a graph.

In this work we are *not* interested in defining new state-of-the-art results, or proposing novel GNN models. We focus on *studying the impact of long-range dependencies*, and identifying a first strategy, which can easily be applied to *any* existing model, to incorporate this information.

**Our Contribution.** In more detail, we assess whether the injection of information on the graph structure that can not be captured by 2 or 3-hop neighbourhoods has a significant impact on the performance of several state-of-the-art GCN models. In this regard, our contributions are fourfold. Firstly, we propose and formalize four different levels of structural information injection. Secondly, we propose a novel and practical regularization strategy, *Random Walk with Restart Regularization* (RWRReg), to inject structural information using random walks with restart, allowing GCNs to leverage long-range dependencies. RWRReg does not require additional operations at inference time, maintains the permutation-invariance of GCN models, and leads to an average $5\%$ increase in accuracy on both node classification, and graph classification. Thirdly, we prove a theoretical result linking random walks with restart and the Weisfeiler-Leman algorithm, providing a theoretical foundation for their use in GCNs. Fourthly, we test how the injection of structural information can impact the performance of 6 different GCN models on node classification, graph classification, and on the task of triangle counting. Results show that current state-of-the-art models lack the ability to extract long-range information, and this is severely affecting their performance.

## 2 INJECTING LONG-RANGE INFORMATION IN GCNS

To test if GCNs are missing on important information that is encoded in the structure of a graph, we inject additional structural information into existing GCN models, and test how the performance of these models changes in several graph related tasks. Intuitively, based on a model's performance when injected with different levels of structural information, we can understand how much information is not captured by GCNs, and if this additional knowledge can improve performance on the considered tasks. In the rest of this section we present the notation used throughout the paper, the four levels of structural information injection that we consider, and an analytical result proving the effectiveness of using information from random walks with restart.

### 2.1 PRELIMINARIES

We use uppercase bold letters for matrices ($\boldsymbol{M}$), and lowercase bold letters for vectors ($\boldsymbol{v}$). We use plain letters with subscript indices to refer to a specific element of a matrix ($M_{i,j}$), or of a vector ($v_i$). We refer to the vector containing the $i$-th row of a matrix with the subscript "$i,:$" ($\boldsymbol{M}_{i,:}$), while we refer to the $i$-th column with the subscript "$:,i$" ($\boldsymbol{M}_{:,i}$).

For a graph $\mathcal{G} = (\mathcal{V}, E)$, where $\mathcal{V} = \{1, .., n\}$ is the set of nodes and $E \subseteq \mathcal{V} \times \mathcal{V}$ is the set of edges, the input is given by a tuple $(\boldsymbol{X}, \boldsymbol{A})$. $\boldsymbol{X}$ is an $n \times d$ matrix where the $i$-th row contains the $d$-dimensional feature vector of the $i$-th node, and $\boldsymbol{A}$ is the $n \times n$ adjacency matrix. For the sake of clarity we restrict our presentation to undirected graphs, but similar concepts can be applied to directed graphs.

### 2.2 STRUCTURAL INFORMATION INJECTION

We consider four different levels of structural information injection, briefly described below. We remark that *not* all the injection strategies presented in this section are made for *practical* use, as the scope of these strategies is to help us understand the importance of missing structural information. In particular, in Section 4 we study the impact of the different types of structural information injection, and hence quantify the information that is not exploited by current GCN models. We then discuss scalability and practicality aspects in Section 5.

**Adjacency Matrix.** We concatenate each node's adjacency matrix row to its feature vector. This explicitly empowers the GCN model with the connectivity of each node, and allows for higher level structural reasoning when considering a neighbourhood (the model will have access to the connectivity of the whole neighbourhood when aggregating messages from neighbouring nodes).

**Random Walk with Restart (RWR) Matrix.** We perform random walks with restart (RWR) (Page et al., 1998) from each node $v$, thus obtaining a $n$-dimensional vector (for each node) that gives a score of how much $v$ is "related" to each other node in the graph. We concatenate this vector of RWR features to each node's feature vector. The choice of RWR is motivated by their capability to capture the relevance between two nodes in a graph (Tong et al., 2006), and by the possibility to modulate the exploration of long-range dependencies by changing the restart probability. Intuitively, if a RWR starting at node $v$ is very likely to visit a node $u$ (e.g. there are multiple paths that connect the two), then there will be a high score in the RWR vector for $v$ at position $u$. This gives the GCN model higher level information about the structure of the graph that goes beyond the 1-hop neighbourhood of each node, and, again, it allows for high level reasoning on neighbourhood connectivity.

**RWR Regularization.** We define a novel regularization term that pushes nodes with mutually high RWR scores to have embeddings that are close to each other (independently of how far they are in the graph). This regularization term encourages the message passing procedure defined by GCNs, that acts on neighbouring nodes, to produce embeddings where pairs of nodes with high RWR score have similar representations. Therefore, the model is encouraged to extract global information, from local communications. The final embeddings are then a combination of local information and long-range information provided by RWR. Let $S$ be the $n \times n$ matrix with the RWR scores. We define the RWRReg (Random Walk with Restart Regularization) loss as follows:

$$\mathcal{L}_{RWRReg} = \sum_{i,j \in V} S_{i,j} ||\boldsymbol{H}_{i,:} - \boldsymbol{H}_{j,:}||^2$$

where $\boldsymbol{H}$ is a matrix of size $n \times d$ containing $d$-dimensional node embeddings that are in between graph convolution layers (see Appendix A for the exact point in which $\boldsymbol{H}$ is considered for each model). With this approach, the loss function used to train the model becomes: $\mathcal{L} = \mathcal{L}_{original} + \lambda \mathcal{L}_{RWRReg}$, where $\mathcal{L}_{original}$ is the original loss function for each model, and $\lambda$ is a balancing term.

**RWR Matrix + RWR Regularization.** We combine the previous two types of structural knowledge injection. The intuition is that it should be easier to enforce the RWRReg by having the additional long-range information provided by the RWR features. We expect this type of information injection to have the highest impact on performance of the models on downstream tasks.

## 2.3 RELATIONSHIP BETWEEN THE 1-WEISFEILER-LEMAN ALGORITHM AND RWRs

In this section we provide analytical evidence that the information from RWR significantly empowers GCNs. In particular, we prove an interesting connection between the 1-Weisfeiler-Leman (1-WL) algorithm and RWR.

The 1-WL algorithm for graph isomorphism testing uses an iterative coloring, or relabeling, scheme, in which all nodes are initially assigned the same label (e.g., the value 1). It then iteratively refines the color of each node by aggregating the multiset of colors in its neighborhood. The final feature representation of a graph is the histogram of resulting node colors. (For a more detailed description of the 1-WL algorithm we refer the reader to Shervashidze et al. (2011).) It is known that there are non-isomorphic graphs that are not distinguishable by the 1-WL algorithm, and that $n$ iterations are enough to distinguish two graphs of $n$ vertices which are distinguishable by the 1-WL algorithm. There is a well known connection (Kipf & Welling, 2017; Xu et al., 2018a) between 1-WL and aggregation-based GCNs, which can be seen as a differentiable approximation of the algorithm. In particular, graphs that can be distinguished in $k$ iterations by the 1-WL algorithm, can be distinguished by *certain* GCNs in $k$ message passing iterations (Morris et al., 2019).

Here, we prove that graphs that are distinguishable by 1-WL in $k$ iterations have different feature representations extracted by RWR of length $k$. Given a graph $G = (V, E)$, we define its *k-step RWR representation* as the set of vectors $\mathbf{r}_v = [r_{v,u_1}, \ldots, r_{v,u_n}]$, $v \in V$, where each entry $r_{v,u}$ describes the probability that a RWR of length $k$ starting in $v$ ends in $u \in V$.

**Proposition 1.** *Let $G_1 = (V_1, E_1)$ and $G_2 = (V_2, E_2)$ be two non-isomorphic graphs for which the 1-WL algorithm terminates with the correct answer after $k$ iterations and starting from the labelling of all 1's. Then the $k$-step RWR representations of $G_1$ and $G_2$ are different.*

The proof can be found in Appendix B. Given that $k$ iterations of the 1-WL algorithm require GCNs of depth $k$ to be performed, but in practice GCNs are limited to depth 2 or 3, the result above shows that RWR can empower GCNs with relevant information that is discarded in practice.

Recent work (Micali & Zhu, 2016) has shown that *anonymous random walks* (i.e., random walks where the global identities of nodes are not known) of fixed length starting at node $u$ are sufficient to reconstruct the local neighborhood within a fixed distance of a node $u$ (Micali & Zhu, 2016). Subsequently, anonymous random walks have been introduced in the context of learning graph representations (Ivanov & Burnaev, 2018). Such results are complementary to ours, since they assume access to the distribution of *entire walks* of a given length, while our RWR representation only stores information on the probability of ending in a given node. In addition, such works do not provide a connection between RWR and 1-WL.

## 3  CHOICE OF MODELS

In order to test the effect of the different levels of structural information injection and to obtain results that are indicative of the whole class of GCN models, our experimental study covers most of the spatial graph convolution techniques. We conceptually identify four different categories from which we select representative models.

**Simple Aggregation Models.**  Such models fall into the *message passing* framework (Gilmer et al., 2017) and utilize a "simple" aggregation strategy, where each node receives messages (e.g. feature vectors) from its neighbours, and uses the received messages to update its embedding vector. As a representative we choose GCN (Kipf & Welling, 2017), one of the fundamental and widely used GNNs models. We also consider GraphSage (Hamilton et al., 2017), as it represents a different aggregation strategy where a set of neighborhood aggregation functions are learned, and a sampling approach is used for defining fixed size neighbourhoods.

**Attention Models.**  Several models have used an attention mechanism in a GNN scenario (Lee et al., 2018a;b; Veličković et al., 2018; Zhang et al., 2018). While they fall into the *message passing* framework, we consider them separately as they employ a more sophisticated aggregation scheme. As a representative we focus on GAT (Veličković et al., 2018), the first to present an attention mechanism over nodes for the aggregation phase, and currently one of the best performing models on several datasets. Furthermore, it can be used in an inductive scenario.

**Pooling Techniques.**  Pooling on graphs is a very challenging task, since it has to take into account that each node might have a different sized neighbourhood. Among the methods that have been proposed for differentiable pooling on graphs (Cangea et al., 2018; Ying et al., 2018b; Diehl et al., 2019; Gao & Ji, 2019; Lee et al., 2019), we choose DiffPool (Ying et al., 2018b) for its strong empirical results. Furthermore, it can learn to dynamically adjust the number of clusters (the number is a hyperparameter, but the network can learn to use fewer clusters if necessary).

**Beyond WL.**  Morris et al. (2019) prove that message-passing GNNs cannot be more powerful than the 1-WL algorithm, and propose $k$-GNNs, which rely on a *subgraph message-passing* mechanism and are proven to be as powerful as the $k$-WL algorithm. Another approach that goes beyond the WL algorithm was proposed by Murphy et al. (2019). Both models are computationally intractable in their initial theoretical formulation, so approximations are needed. As representative we choose $k$-GNNs, to test if subgraph message-passing is affected by additional structural information.

## 4  EVALUATION OF THE INJECTION OF STRUCTURAL INFORMATION

We now present our framework for evaluating the effects of the injection of structural information into GNNs, and the results of our experiments. We consider one *transductive* task (node classification) and two *inductive* tasks (graph classification, and triangle counting), and we further study the impact of the restart probability of RWR on the results.

We use each architecture for the task that better suits its design: GCN, GraphSage, and GAT for node classification, and DiffPool and $k$-GNN for graph classification. We add an adapted version of

Table 1: Node classification accuracy results of different models with added Adjacency matrix features (AD), RWR features (RWR), RWR Regularization (RWRReg), and RWR features + RWR Regularization (RWR+RWRReg).

| Model | Structural | Dataset | | |
|---|---|---|---|---|
| | Information | Cora | Pubmed | Citeseer |
| GCN | none | $0.799 \pm 0.029$ | $0.776 \pm 0.022$ | $0.663 \pm 0.095$ |
| | AD | $0.806 \pm 0.035$ | $0.779 \pm 0.070$ | $0.653 \pm 0.104$ |
| | RWR | $0.817 \pm 0.025$ | $0.782 \pm 0.042$ | $0.665 \pm 0.098$ |
| | RWRReg | $\mathbf{0.861 \pm 0.025}$ | $0.799 \pm 0.034$ | $0.686 \pm 0.096$ |
| | RWR+RWRReg | $0.842 \pm 0.026$ | $\mathbf{0.811 \pm 0.037}$ | $\mathbf{0.690 \pm 0.102}$ |
| GraphSage | none | $0.806 \pm 0.017$ | $0.807 \pm 0.016$ | $0.681 \pm 0.021$ |
| | AD | $0.803 \pm 0.014$ | $0.803 \pm 0.013$ | $0.688 \pm 0.020$ |
| | RWR | $0.816 \pm 0.014$ | $0.807 \pm 0.015$ | $0.693 \pm 0.019$ |
| | RWRReg | $\mathbf{0.841 \pm 0.016}$ | $0.818 \pm 0.017$ | $0.721 \pm 0.021$ |
| | RWR+RWRReg | $0.837 \pm 0.015$ | $\mathbf{0.820 \pm 0.010}$ | $\mathbf{0.728 \pm 0.020}$ |
| GAT | none | $0.815 \pm 0.021$ | $0.804 \pm 0.011$ | $0.664 \pm 0.008$ |
| | AD | $0.823 \pm 0.019$ | $0.796 \pm 0.014$ | $0.672 \pm 0.017$ |
| | RWR | $0.833 \pm 0.020$ | $0.811 \pm 0.009$ | $0.686 \pm 0.009$ |
| | RWRReg | $0.824 \pm 0.022$ | $0.811 \pm 0.013$ | $\mathbf{0.702 \pm 0.013}$ |
| | RWR+RWRReg | $\mathbf{0.848 \pm 0.019}$ | $\mathbf{0.828 \pm 0.010}$ | $0.701 \pm 0.011$ |

GCN for graph classification, as a common strategy for this task is to deploy a node-level GNN, and then apply a *readout* function to combine node embeddings into a global graph embedding vector.

With regards to datasets, for node classification we considered the three most used benchmarking datasets in literature: Cora, Citeseer, and Pubmed (Sen et al., 2008). Analogously, for graph classification we chose three frequently used datasets: ENZYMES, PROTEINS, and D&D (Kersting et al., 2016). Dataset statistics can be found in Appendix C.

For all the considered models we take the hyperparameters from the implementations released by the authors. The only parameter tuned using the validation set is the balancing term $\lambda$ when RWRReg is applied. We found that the RWRReg loss tends to be larger than the Cross Entropy loss for prediction, and the best values for $\lambda$ lie in the range $[10^{-9}, 10^{-6}]$. For all the RWR-based techniques we used a restart probability of $0.15$[1]. (The effects of different restart probabilities are explored below.) Detailed information on our implementations can be found in Appendix A [2].

**Node Classification.** For each dataset we follow the approach that has been widely adopted in literature: we take 20 labeled nodes per class as training set, 500 nodes as validation set, and 1000 nodes for testing. Most authors have used the train/validation/test split defined by Yang et al. (2016). Since we want to test the general effect of the injection of structural information, we differ from this approach and we do not rely on a single split. We perform 100 runs, where at each run we randomly sample 20 nodes per class for training, 500 random nodes for validation, and 1000 random nodes for testing. We then report mean and standard deviation for the accuracy on the test set over these 100 runs.

Results are summarized in Table 1, where we observe that the simple addition of RWR features to the feature vector of each node is sufficient to give a performance gain (up to 2%). The RWRReg term then significantly increments the gain (up to **7.5%**), showing that even for the task of node classification structural information and long-range information are important, confirming that only looking at neighbours and close nodes is not enough.

**Graph Classification.** Following the approach from Ying et al. (2018b) and Morris et al. (2019) we use 10-fold cross validation, and report mean and standard deviation of the accuracy on graph classification. Results are summarized in Table 2. The performance gains given by the injection of structural information are even more apparent than for the node classification task. Intuitively, the

---

[1] We use 0.15 as it is a common default value used in many papers and software libraries.

[2] Source code is provided as Supplementary Material and will be made publicly available upon acceptance.

Table 2: Graph classification accuracy results of different models with added Adjacency matrix features (AD), RWR features (RWR), RWR Regularization (RWRReg), and RWR features + RWR Regularization (RWR+RWRReg).

| Model | Structural Information | Dataset | | |
| --- | --- | --- | --- | --- |
| | | ENZYMES | D&D | PROTEINS |
| GCN | none | $0.570 \pm 0.052$ | $0.755 \pm 0.028$ | $0.740 \pm 0.035$ |
| | AD | $0.591 \pm 0.076$ | $0.779 \pm 0.022$ | $0.775 \pm 0.042$ |
| | RWR | $0.584 \pm 0.055$ | $0.775 \pm 0.023$ | $0.784 \pm 0.034$ |
| | RWRReg | $\mathbf{0.621 \pm 0.041}$ | $0.786 \pm 0.024$ | $0.785 \pm 0.036$ |
| | RWR+RWRReg | $0.616 \pm 0.065$ | $\mathbf{0.790 \pm 0.023}$ | $\mathbf{0.795 \pm 0.032}$ |
| DiffPool | none | $0.661 \pm 0.031$ | $0.793 \pm 0.022$ | $0.813 \pm 0.017$ |
| | AD | $0.711 \pm 0.027$ | $0.837 \pm 0.020$ | $0.821 \pm 0.039$ |
| | RWR | $0.687 \pm 0.025$ | $0.824 \pm 0.028$ | $0.783 \pm 0.043$ |
| | RWRReg | $\mathbf{0.733 \pm 0.032}$ | $0.822 \pm 0.025$ | $0.820 \pm 0.038$ |
| | RWR+RWRReg | $0.721 \pm 0.039$ | $\mathbf{0.840 \pm 0.024}$ | $\mathbf{0.834 \pm 0.038}$ |
| $k$-GNN | none | $0.515 \pm 0.111$ | $0.756 \pm 0.021$ | $0.763 \pm 0.043$ |
| | AD | $0.572 \pm 0.063$ | $0.778 \pm 0.020$ | $0.751 \pm 0.034$ |
| | RWR | $0.573 \pm 0.077$ | $\mathbf{0.794 \pm 0.022}$ | $0.781 \pm 0.028$ |
| | RWRReg | $\mathbf{0.582 \pm 0.075}$ | $0.787 \pm 0.022$ | $0.780 \pm 0.028$ |
| | RWR+RWRReg | $0.571 \pm 0.080$ | $0.786 \pm 0.021$ | $\mathbf{0.785 \pm 0.026}$ |

structure of the nodes in a graph is fundamental for distinguishing different graphs. Most notably, the addition of the adjacency features is sufficient to give a large performance boost (up to **11%**).

Surprisingly, models like DiffPool and $k$-GNN show an important difference in accuracy (up to **10%**) when there is injection of structural information, meaning that even the most advanced methods suffer from the inability to properly exploit all the structural information encoded in a graph.

**Impact of RWR Restart Probability.** We tested how performance change with different restart probabilities. Intuitively, higher restart probabilities might put too much focus on close nodes, while lower probabilities may focus too much on nodes that are "central" in the graph structure, with fewer differences in the RWR features between nodes. Figure 1 (a) summarises how the accuracy on node classification changes with different restart probabilities. Results for graph classification are shown in Figure 1 (b). In accordance to our intuition, higher restart probabilities focus on close nodes (and less on distant nodes), and produce lower accuracies. Furthermore, we notice how injecting RWR information is never detrimental to the performance of the model without any injection.

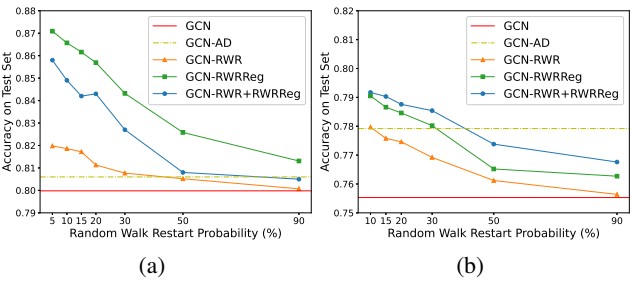

Figure 1: Accuracy on Cora (a), and on D&D (b), of GCN without and with the injection of structural information, and for different restart probabilities of RWR.

**Counting Triangles.** The TRIANGLES dataset Knyazev et al. (2019) is composed of randomly generated graphs, where the task is to count the number of triangles contained in each graph. This is a hard task for GNNs as the aggregation of neighbouring node's features with permutation invariant functions does not allow the model to explicitly access to structural information. The TRIANGLES dataset has a test set with 10'000 graphs, of which half are similar in size to the ones in the training and validation sets (4-25 nodes), and half are bigger (up to 100 nodes). This permits an evaluation of the generalization capabilities to graphs of unseen sizes.

For this regression task we use a three layer GCN, and we minimize the Mean Squared Error (MSE) loss (more details can be found in Appendix A). Table 3 presents MSE results on the test dataset as a whole and on the two splits separately. We see that the addition of RWR features and of RWRReg provides significant benefits (up to **19%** improvements), specially when the model has to generalize to graphs of unseen sizes, while the addition of adjacency features leads to overfitting.

## 5 PRACTICAL RWR REGULARIZATION

As shown in Section 4, the addition of RWR features as node features coupled with RWRReg provides a significant improvement of the accuracy on all considered tasks. However, these benefits come at a high cost: adding RWR features increases the input size of $n \times n$ elements (which is prohibitive for large graphs), and RWRReg requires the computation of an additional loss term (and the storage of the RWR matrix) during training. Furthermore, all the considered models have a weight matrix at each layer that depends on the feature

Table 3: Mean Squared Error (MSE) of GCN with different levels of structural information injection on the TRI-ANGLES test set.

| Model | TRIANGLES Test Set | | |
|---|---|---|---|
| | Global | Small | Large |
| GCN | 2.290 | 1.311 | 3.608 |
| GCN-AD | 4.746 | 1.162 | 5.971 |
| GCN-RWR | 2.044 | **1.101** | 2.988 |
| GCN-RWRReg | 2.187 | 1.282 | 3.014 |
| GCN-RWR+RWRReg | **2.029** | 1.166 | **2.893** |

dimension, which means we are also increasing the number of parameters at the first layer by $n \times d^{(1)}$ (where $d^{(1)}$ is the dimension of the feature vector for each node after the first GCN layer). In this section we propose a practical way to take advantage of the injection of structural information without increasing the number of parameters, and controlling the memory consumption during training.

The results in Section 4 show that the sole addition of the RWRReg term increases the performance of the considered models by more than **5%**. Furthermore, RWRReg does not increase the size of the input or the number of parameters (as it does **not** add any feature to the node's feature vectors), does not require additional operations at inference time, and maintains the permutation invariance of GCN models. Therefore, RWRReg alone is a very practical tool that significantly improves the quality of GCN models. However, when dealing with very large graphs, keeping in memory the RWR matrix to compute RWRReg during training might be too expensive. We then explore how the sparsification of this matrix affects the resulting model. In particular, we apply a *top-K* strategy: for each node, we only keep the $K$ highest RWR weights. Figure 2 shows how different values of $K$ impact performance on node classification (which usually is the task with the largest graphs). We can see that the addition of the RWRReg term is always beneficial. Furthermore, by taking the *top-$\frac{n}{2}$*, we can reduce the number of entries in the RWR matrix of $\frac{n^2}{2}$ elements, while still obtaining an average **3.2%** increment on the accuracy of the model. This strategy then allows the selection of the value of $K$ that best suits the available memory, while still obtaining a high performing model (better than GCN without structural information injection).

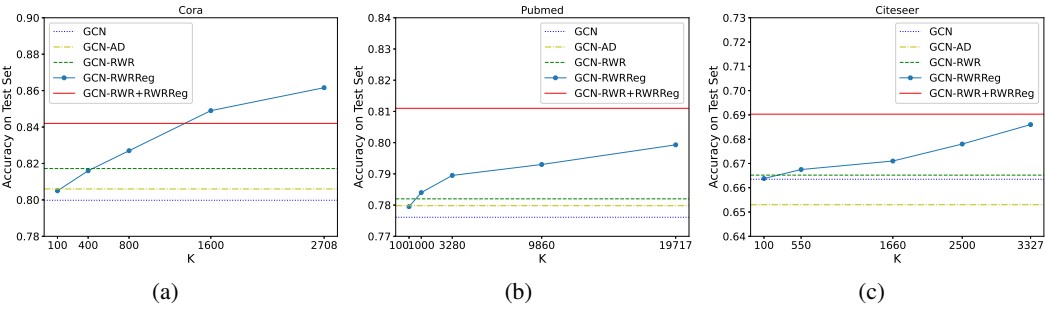

Figure 2: Performance of GCN-RWRReg on node classification when trained using *Top-K* sparsification of the RWR matrix.

## 6 RELATED WORK

The field of GNNs has become extremely vast, for a thorough review we refer the reader to the latest survey on the subject (Wu et al., 2019). To the best of our knowledge there are no studies that test if additional structural information can significantly impact GCNs, and there has been very few interest in long-range dependencies between nodes. However, there are some works that are conceptually related to our approach.

Klicpera et al. (2019b) use RWR to create a new (weighted) adjacency matrix where message passing is performed. While this can enable long-range communication, it is impractical for inductive scenarios, as the RWR matrix needs to be calculated for each new graph. In contrast, our RWRReg method only uses the RWR matrix in the training phase, and does not require any additional operation at inference time. Other works have used random walks with GCNs in different ways. Li et al. (2018) use random walks in a co-training scenario to add new nodes for the GCN's training set. Ying et al. (2018a) and Zhang et al. (2019) use random walks to define aggregation neighbourhoods that are not confined to a fixed distance. Abu-El-Haija et al. (2018) and Abu-El-Haija et al. (2019) use powers of the adjacency matrix, which can be considered as random walk statistics, to define neighbourhoods of different scales. Zhuang & Ma (2018) use random walks to define the positive pointwise mutual information (PPMI) matrix and then use it in place of the adjacency matrix in the GCN formulation. Klicpera et al. (2019a) use a diffusion strategy based on RWR instead of aggregating information from neighbours. This last work has recently been extended by Bojchevski et al. (2020) to scale to large graphs using RWRs to sample neighbourhoods. We remark how all the aforementioned papers focus on creating smart or extended neighbourhoods which are then used for node aggregation, while we show that node aggregation (or message-passing) without additional information (e.g., RWR features or RWR-based regularization) is not capable of fully exploiting structural graph information.

Pei et al. (2020) propose a strategy to insert long-range dependencies information in GCNs by performing aggregation between neighbours in a latent space obtained with some classical node embedding techniques, but it is limited to transductive tasks. Our method can be easily applied to any existing GCN architecture, and works also on inductive tasks. Gao et al. (2019), and Jiang & Lin (2018) use regularization techniques to enforce that the embeddings of neighbouring nodes should be close to each other. The first uses Conditional Random Fields, while the second uses a regularization term based on the graph Laplacian. Both approaches only focus on 1-hop neighbours and do not take long-range dependencies into account.

With regards to the study of the capabilities and weaknesses of GNNs, Li et al. (2018) and Xu et al. (2018b) study the over-smoothing problem that appears in Deep-GCN architectures, while Xu et al. (2018a) and Morris et al. (2019) characterize the relation to the Weisfeiler-Leman algorithm. Other works have expressed the similarity with distributed computing (Sato et al., 2019; Loukas, 2020), and the alignment with particular algorithmic structures (Xu et al., 2020). These important contributions have advanced our understanding of the capabilities of GNNs, but they do not quantify the impact of additional structural information.

Our approach relies on the computation of the RWR matrix for training the model. When dealing with large graphs, there is a vast literature on fast approximations of RWR scores (Andersen et al., 2006; Tong et al., 2006; Bahmani et al., 2010; Lofgren, 2015; Wei et al., 2018; Wang et al., 2019).

## 7 CONCLUSIONS

In this work we showed that state-of-the-art GCN models ignore relevant information regarding node and graph similarity that is revealed by long distance relations among nodes. We describe four ways to inject such information in several models, and empirically show that the performance of all models significantly improve when such information is used. We then propose a novel regularization technique based on RWR, which leads to an average improvement of $5\%$ on all models. Our experimental results are supported by a novel connection between RWR and the 1-Weisfeiler-Leman algorithm, which proves that RWR encode long-range relations that are not captured by considering only neighbours at distance at most 2 or 3, as it is common practice in GCNs. Based on our results, there are several interesting directions for future research, including the design of GCN architectures that directly capture long distance relations.

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

## A   MODEL IMPLEMENTATION DETAILS

We present here a detailed description of the implementations of the models we use in our experimental section. Whenever possible, we started from the official implementation of the authors of each model. Table 4 contains links to the implementations we used as starting point for the code for our experiments. Our code is available as supplementary material and will be made publicly available after acceptance.

Table 4: Starting model implementations.

| Model | Implementation |
|---|---|
| GCN *(for node classification)* | github.com/tkipf/pygcn |
| GCN *(for graph classification)* GCN *(for triangle counting)* | github.com/bknyaz/graph_nn |
| GraphSage | github.com/williamleif/graphsage-simple |
| GAT | github.com/Diego999/pyGAT |
| DiffPool | github.com/RexYing/diffpool |
| $k$-GNN | github.com/chrsmrrs/k-gnn |

**Training Details.**   With regards to the training procedure we have that all models are trained with early stopping on the validation set (stopping the training if the validation loss doesn't decrease for a certain amount of epochs), and unless explicitly specified, we use Cross Entropy as loss function for all the classification tasks.

For the task of graph classification we zero-pad the feature vectors of each node to make them all the same length when we inject structural information into the node feature vectors.

For the task of triangle counting we follow Knyazev et al. (2019) and use the one-hot representation of node degrees as node feature vectors to impose some structural information in the network.

**Computing Infrastructure.**   The experiments were run on a GPU cluster with 7 Nvidia 1080Ti, and on a CPU cluster (when the memory consumption was too big to fit in the GPUs) equipped with 8 cpus 12-Core Intel Xeon Gold 5118 @2.30GHz, with 1.5Tb of RAM.

In the rest of this Section we go through each model used in our experiments, specifying architecture, hyperparameters, and the position of the node embeddings used for RWRReg.

### A.1   GCN *(node classification)*

We use a two layer architecture. The first layer outputs a 16-dimensional embedding vector for each node, and passes it through a ReLu activation, before applying dropout Srivastava et al. (2014), with probability 0.5. The second layer outputs a $c$-dimensional embedding vector for each node, where $c$ is the number of output classes and these vectors are passed through *Softmax* to get the output probabilities for each class. An additional L2-loss is added with a balancing term of 0.0005. The model is trained using the Adam optimizer Kingma & Ba (2015) with a learning rate of 0.01.

We apply the RWRReg on the 16-dimensional node embeddings after the first layer.

### A.2   GCN *(graph classification)*

We first have two GCN layers, each one generating a 128-dimensional embedding vector for each node. Then we apply *max*-pooling on the features of the nodes and pass the pooled 128-dimensional vector to a two-layer feed-forward neural network with 256 neurons at the first layer and $c$ at the last one, where $c$ is the number of output classes. A ReLu activation is applied in between the two feed-forward layers, and *Softmax* is applied after the last layer. Dropout Srivastava et al. (2014) is applied in between the last GCN layer and the feed-forward layer, and in between the feedforward layers (after ReLu), in both cases with probability of 0.1. The model is trained using the Adam optimizer Kingma & Ba (2015) with a learning rate of 0.0005.

We apply the RWRReg on the 128-dimensional node embeddings after the last GCN layer.

### A.3   GCN *(counting triangles)*

We first have three GCN layers, each one generating a 64-dimensional embedding vector for each node. Then we apply *max*-pooling on the features of the nodes and pass the pooled 64-dimensional vector to a one-layer feed-forward neural network with one neuron. Dropout Srivastava et al. (2014) is applied in between the last GCN layer and the feed-forward layer with probability of 0.1. The

model is trained by minimizing the mean squared error (MSE) and is optimized using the Adam optimizer Kingma & Ba (2015) with a learning rate of 0.005.

We apply the RWRReg on the 64-dimensional node embeddings after the last GCN layer.

## A.4    GRAPHSAGE

We use a two layer architecture. For Cora we sample 5 nodes per-neighbourhood at the first layer and 5 at the second, while on the other datasets we sample 10 nodes per-neighbourhood at the first layer and 25 at the second. Both layers are composed of *mean-aggregators* (i.e., we take the mean of the feature vectors of the nodes in the sampled neighbourhood) that output a 128-dimensional embedding vector per node. After the second layer these embeddings are multiplied by a learnable matrix with size $128 \times c$, where $c$ is the number of output classes, giving thus a $c$-dimensional vector per-node. These vectors are passed through *Softmax* to get the output probabilities for each class. The model is optimized using Stochastic Gradient Descent with a learning rate of 0.7.

We apply the RWRReg on the 128-dimensional node embeddings after the second aggregation layer.

## A.5    GAT

We use a two layer architecture. The first layer uses an 8-headed attention mechanism that outputs an 8-dimensional embedding vector per-node. LeakyReLu is set with slope $\alpha = 0.2$. Dropout Srivastava et al. (2014) (with probability of 0.6) is applied after both layers. The second layer outputs a $c$-dimensional vector for each node, where $c$ is the number of classes, and before passing each vector through *Softmax* to obtain the output predictions, the vectors are passed through an Elu activation Clevert et al. (2016). An additional L2-loss is added with a balancing term of 0.0005. The model is optimized using Adam Kingma & Ba (2015) with a learning rate of 0.005.

We apply the RWRReg on the 8-dimensional node embeddings after the first attention layer. A particular note needs to be made for the training of GATs: we found that naively implementing the RWRReg term on the node embeddings in between two layers brings to an exploding loss as the RWRReg term grows exponentially at each epoch. We believe this happens because the attention mechanism in GATs allows the network to infer that certain close nodes, even 1-hop neighbours, might not be important to a specific node and so they shouldn't be embedded close to each other. This clearly goes in contrast with the RWRReg loss, since 1-hop neighbours always have a high score. We solved this issue by using the attention weights to scale the RWR coefficients at each epoch (we make sure that gradients are not calculated for this operation as we only use them for scaling). This way the RWRReg penalizations are in accordance with the attention mechanism, and are still encoding long-range dependencies.

## A.6    DIFFPOOL

We use a 1-pooling architecture. The initial node feature matrix is passed through two (one to obtain the assignment matrix and one for node embeddings) 3-layer GCN, where each layer outputs a 20-dimensional vector per-node. Pooling is then applied, where the number of clusters is set as 10% of the number of nodes in the graph, and then another 3-layer GCN is applied to the pooled node features. Batch normalization Ioffe & Szegedy (2015) is added in between every GCN layer. The final graph embedding is passed through a 2-layer MLP with a final *Softmax* activation. An additional L2-loss is added with a balancing term of $10^{-7}$, together with two pooling-specific losses. The first enforces the intuition that nodes that are close to each other should be pooled together and is defined as: $\mathcal{L}_{LP} = \|\boldsymbol{A}^{(l)}, \boldsymbol{S}^{(l)\intercal}\boldsymbol{S}^{(l)}\|_F$, where $\| \cdot \|_F$ is the Frobenius norm, and $\boldsymbol{S}^{(l)}$ is the assignment matrix at layer $l$. The second one encourages the cluster assignment to be close to a one-hot vector, and is defined as: $\mathcal{L}_E = \frac{1}{n}\sum_{i=1}^{n} H(\boldsymbol{S}_{i,:})$, where $H$ is the entropy function. However, in the implementation available online, the authors do not make use of these additional losses. We follow the latter implementation. The model is optimized using Adam Kingma & Ba (2015) with a learning rate of 0.001.

We apply the RWRReg on the 20-dimensional node embeddings after the first 3-layer GCN (before pooling). We tried applying it also after pooling on the coarsened graph, but the fact that this graph could change during training yields to poor results.

### A.7 $k$-GNN

We use the hierarchical 1-2-3-GNN architecture (which is the one showing the highest empirical results). First a 1-GNN is applied to obtain node embeddings, then these embeddings are used as initial values for the 2 GNN (1-2-GNN). The embeddings of the 2-GNN are then used as initial values for the 3-GNN (1-2-3-GNN). The 1-GNN applies 3 graph convolutions, while 2-GNN and the 3-GNN apply 2 graph convolutions. Each convolution outputs a 64-dimensional vector and is followed by an Elu activation Clevert et al. (2016). For each $k$, node features are then globally averaged and the final vectors are concatenated and passed through a three layer MLP. The first layer outputs a 64-dimensional vector, while the second outputs a 32-dimensional vector, and the third outputs a $c$-dimensional vector, where $c$ is the number of output classes. To obtain the final output probabilities for each class, *log(Softmax)* is applied, and the negative log likelihood is used as loss function. After the first and the second MLP layers an Elu activation Clevert et al. (2016) is applied, furthermore, after the first MLP layer dropout Srivastava et al. (2014) is applied with probability 0.5. The model is optimized using Adam Kingma & Ba (2015) with a learning rate of 0.01, and a decaying learning rate schedule based on validation results (with minimum value of $10^{-5}$).

We apply the RWRReg on the 64-dimensional node embeddings after the 1-GNN. We were not able to apply it also after the 2-GNN and the 3-GNN, as it would cause out-of-memory issues with our computing resources.

## B  PROOF OF PROPOSITION 1

Given a graph $G = (V, E)$, we define its *k-step RWR representation* as the set of vectors $\mathbf{r}_v = [r_{v,u_1}, \ldots, r_{v,u_n}]$, $v \in V$, where each entry $r_{v,u}$ describes the probability that a RWR of length $k$ starting in $v$ ends in $u$.

**Proposition 2.** *Let $G_1 = (V_1, E_1)$ and $G_2 = (V_2, E_2)$ be two non-isomorphic graphs for which the 1-WL algorithm terminates with the correct answer after $k$ iterations and starting from the labelling of all 1's. Then the $k$-step RWR representations of $G_1$ and $G_2$ are different.*

*Proof.* Consider the WL algorithm with initial labeling given by all 1's. It's easy to see that i) after $k$ iterations the label of a node $v$ corresponds to the information regarding the degree distribution of the neighborhood of distance $\leq k$ from $v$ and ii) in iteration $i \leq k$, the degrees of nodes at distance $i$ from $v$ are included in the label of $v$. In fact, after the first iteration, two nodes have the same colour if they have the same degree, as the colour of each node is given by the multiset of the colours of its neighbours (and we start with initial labeling given by all 1's). After the second colour refinement iteration two nodes have the same colour if they had the same colour after the first iteration (i.e. have the same degree), and the multisets containing the colours (degrees) of their neighbours are the same. In general, after the $k$-th iteration, two nodes have the same colour if they had the same colour in iteration $k - 1$, and the multiset containing the degrees of the neighbours at distance $k$ is the same for the two nodes. Hence, two nodes that have different colours after a certain iteration, will have different colours in all the successive iterations. Furthermore, the colour after the $k$-th iteration depends on the colour at the previous iteration (which "encodes" the distribution of degree of neighbours up to distance $k-1$ included), and the multiset of the degrees of neighbours at distance $k$.

Given two non-isomorphic graphs $G_1$ and $G_2$, if the WL algorithm terminates with the correct answer starting from the all 1's labelling in $k$ iterations, it means that there is no *matching* between vertices in $V_1$ and vertices in $V_2$ such that matched vertices have the same degree distribution for neighborhoods at distance exactly $k$. Equivalently, any matching $M$ that minimizes the number of matched vertices with different degree distribution has at least one such pair. Now consider one such matching $M$, and let $v \in V_1$ and $w \in V_2$ be vertices matched in $M$ with different degree distributions for neighborhoods at distance exactly $k$. Since $v$ and $w$ have different degree distributions at distance $k$, the number of choices for paths of length $k$ starting from $v$ and $w$ must be different (since the number of choices for the $k$-th edge on the path is different). Therefore, there must be at least a node $u \in V_1$ and a node $z \in V_2$ that are matched by $M$ but for which the number of paths of length $k$ from $v$ to $u$ is different from the number of paths of length $k$ from $w$ to $z$. Since $r_{v,u}$ is proportional

Table 5: Node classification dataset statistics.

| Dataset | Nodes | Edges | Classes | Features | Label Rate |
|---------|-------|-------|---------|----------|------------|
| Cora | 2708 | 5429 | 7 | 1433 | 0.052 |
| Pubmed | 19717 | 44338 | 3 | 500 | 0.003 |
| Citeseer | 3327 | 4732 | 6 | 3703 | 0.036 |

Table 6: Graph classification and triangle counting dataset statistics.

| Dataset | Graphs | Classes | Avg. # Nodes | Avg. # Edges |
|---------|--------|---------|--------------|--------------|
| ENZYMES | 600 | 6 | 32.63 | 62.14 |
| D&D | 1178 | 2 | 284.32 | 715.66 |
| PROTEINS | 1113 | 2 | 39.1 | 72.82 |
| TRIANGLES | 45000 | 10 | 20.85 | 32.74 |

to the number of paths of length $k$ from $v$ to $u$, we have that $r_{v,u} \neq r_{w,z}$, that is $\mathbf{r}_v \neq \mathbf{r}_w$. Thus, the *k-step RWR representation* of $G_1$ and $G_2$ are different. $\qquad\square$

## C  DATASETS

We briefly present here some additional details about the datasets used for our experimental section. Table 5 summarizes the datasets for node classification, while Table 6 presents information about the datasets for graph classification and triangle counting. The node classification datasets are available at `https://linqs.soe.ucsc.edu/data`, while the graph classification and the triangle counting at `https://ls11-www.cs.tu-dortmund.de/staff/morris/graphkerneldatasets`.

## D  FAST IMPLEMENTATION OF THE RANDOM WALK WITH RESTART REGULARIZATION

Let $\boldsymbol{H}$ be the matrix containing the node embeddings, and $\boldsymbol{S}$ be the matrix with the RWR statistics. We are interested in the following quantity

$$\mathcal{L}_{RWRReg} = \sum_{i,j} S_{i,j} ||\boldsymbol{H}_{i,:} - \boldsymbol{H}_{j,:}||^2$$

To calculate it in a fast way (specially when using GPUs) we use the following procedure. Let us first define the following matrices:

$$\hat{\boldsymbol{S}} = n \times n \text{ symmetric matrix with } \hat{S}_{i,j} = \begin{cases} S_{i,j} + S_{j,i} & \text{for } i \neq j \\ S_{i,j} & \text{for } i = j \end{cases}$$

$$\boldsymbol{D} = n \times n \text{ diagonal matrix with } D_{i,i} = \sum_j \hat{S}_{i,j}$$

$$\boldsymbol{\Delta} = \boldsymbol{D} - \hat{\boldsymbol{S}}$$

Where we are allowed to make $\boldsymbol{S}$ symmetric because $||\boldsymbol{H}_{i,:} - \boldsymbol{H}_{j,:}|| = ||\boldsymbol{H}_{j,:} - \boldsymbol{H}_{i,:}||, \forall i, j$. We then have

$$\mathcal{L}_{RWRReg} = \sum_{i,j} S_{i,j} ||\boldsymbol{H}_{i,:} - \boldsymbol{H}_{j,:}||^2 = \sum_i \boldsymbol{H}_{:,i}^\mathsf{T} \boldsymbol{\Delta} \boldsymbol{H}_{:,i} = Tr\left(\boldsymbol{H}^\mathsf{T} \boldsymbol{\Delta} \boldsymbol{H}\right)$$

Where $Tr(\cdot)$ is the trace of the matrix. Note that $\boldsymbol{H}_{:,i}^\mathsf{T}$ is the $i$-th column of $\boldsymbol{H}$, transposed, so its size is $1 \times n$.

## E    EMPIRICAL ANALYSIS OF THE RANDOM WALK WITH RESTART MATRIX

We now analyse the RWR matrix to justify the use of RWR for the encoding of long range dependencies, and other important structural information. We consider the three node classification datasets (see Section 4 of the paper), as this is the task with the largest input graphs, and hence where this kind of information seems more relevant.

We first consider the distribution of the RWR[3] weights at different distances from a given node. In particular, for each node, we take the sum of the weights assigned to the 1-hop neighbours, the 2-hop neighbours, and so on. We then take the average, over all nodes, of the sum of the RWR weights at each hop. We discard nodes that belong to connected components with diameter $\leq 4$, and we only plot the values for the distances that have

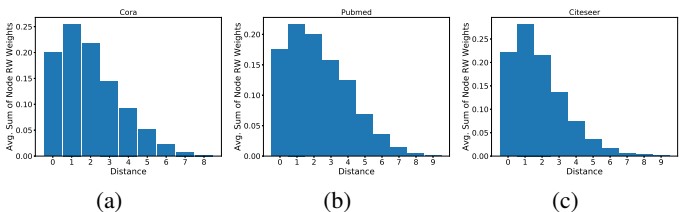

(a)                    (b)                    (c)

Figure 3: Average distribution of the RWR weights at different distances for the node classification datasets. Distance zero indicates the weight that a node assigns to itself.

an average sum of weights higher than 0.001. Plots are shown in Figure 3. We notice that the RWR matrix contains information that goes beyond the immediate neighbourhood of a node. In fact, we see that approximately 90% of the weights are contained within the 6-hop neighbourhood, with a significant portion that is not contained in the 2-hop neighbourhood usually accessed by GCN-like models.

Next we analyse if RWR capture some non-trivial relationships between nodes. In particular, we investigate if there are nodes that are far from the starting node, but receive a higher weight than some closer nodes. To quantify this property we use the Kendall Tau-b[4] measure (Kendall (1945)). In more detail, for each node $v$ we consider the sequence $rw^{(v)}$ where the $i$-th element is the weight that the RWR from node $v$ has assigned to node $i$: $rw^{(v)}[i] = S_{v,i}$. We then define the sequence $drw^{(v)}$ such that $drw^{(v)}[j] = dist(v, f_{sort\_weights}(j, rw^{(v)}))$, where $dist(x, y)$ is the shortest path distance between node $x$ and node $y$, and $f_{sort\_weights}(j, rw^{(v)})$ is the node with the $j$-th highest RWR weight in $rw^{(v)}$. Intuitively, if the RWR matrix isn't capable of capturing non-trivial relationship we would have that $drw^{(v)}$ is a sorted

Table 7: Average and standard deviation, over all nodes, of Kendall Tau-b values measuring the non-trivial relationships between nodes captured by the RWR weights.

| Dataset | Average Kendall Tau-b |
|---------|----------------------|
| Cora | $0.729 \pm 0.082$ |
| Pubmed | $0.631 \pm 0.057$ |
| Citeseer | $0.722 \pm 0.171$ |

list (with repetitions). By comparing $drw^{(v)}$ with its sorted version with the Kendall Tau-b rank, we obtain a value between 1 and $-1$ where 1 means that the two sequences are identical, and $-1$ means that one is the reverse of the other. Table 7 presents the results, averaged over all nodes, on the node classification datasets. These results show that while there is a strong relation between the information provided by RWR and the distance between nodes, there is information in the RWR that is not captured by shortest path distances.

As an example of the non-trivial relationships encoded by RWR, Figure 4 presents a $drw^{(v)}$ sequence taken from a node in Cora. This sequence obtains a Kendall Tau-b value of 0.591. We can observe that the nodes at distance 1 are the nodes with the highest weights, however, for distances greater than 1, we already have some non-trivial relationships. In fact, we observe some nodes at distance 3 that receive a larger weight than nodes at distance 2. There are many other interesting non-trivial relationships, for example we notice that some nodes at distance 7, and some at distance 11, obtain a higher weight than some nodes at distance 5.

---

[3]We consider RWR, with a restart probability of 0.15, as done for the experimental evaluation of our proposed technique.

[4]We use the Tau-b version because the elements in the sequences we analyze are not all distinct.

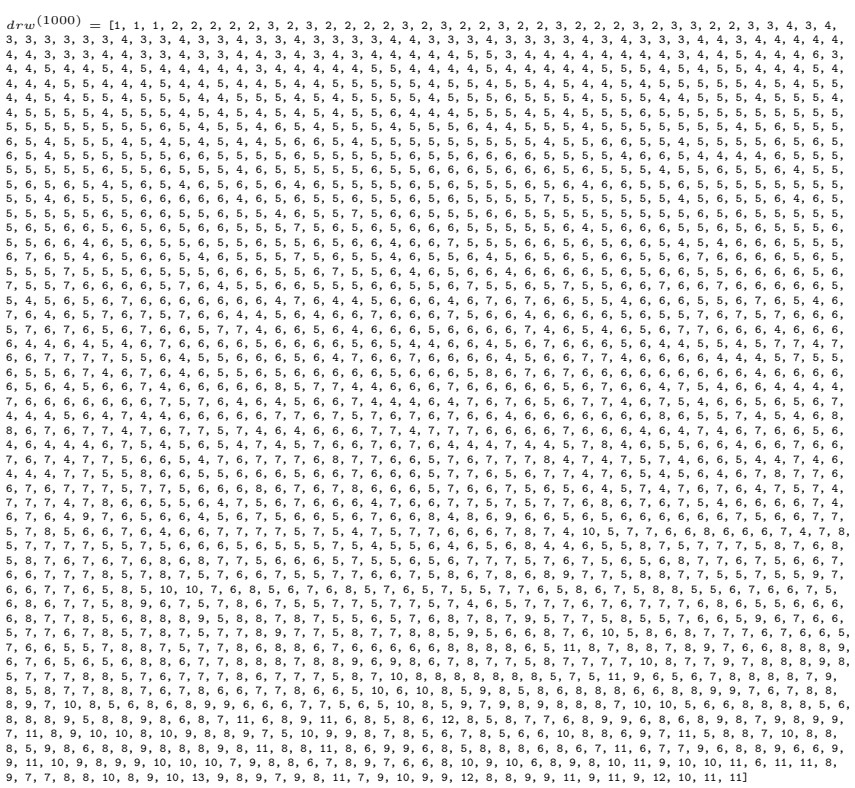

Figure 4: $drw^{(v)}$ sequence for the 1000-th node in Cora.

