# OpenReview forum: "Are Graph Convolutional Networks Fully Exploiting the Graph Structure?"
_ICLR.cc/2021/Conference — Reject_

### Official Review · AnonReviewer2 · 2020-10-27
**This paper proposes to use k-step RWR to capture long-range structural information as a part of node features, so as to overcome the limitation of GCN that it cannot infer long-range dependencies. However, the methodology is incremental and cannot easily be applied to real-world large-scale networks. In summary, the originality and significance are limited.**

**Rating:** 4
**Confidence:** 4

**Review:**

This paper proposes to use k-step RWR to capture long-range structural information as a part of node features, so as to overcome the limitation of GCN that it cannot infer long-range dependencies. However, the methodology is incremental and cannot easily be applied to real-world large-scale networks. In summary, the originality and significance are limited (please see detailed comments in the following).



Pros:

1. On the premise that GIN [5] has proved the connection between GNNs and WL test, similarly, this work proves the connection between k-step RWR and WL test.

2. The experimental results seem promising. The model implementation details in the appendix are extensive, which facilitates good reproducibility.

3. This paper is written well and has good clarity.



Cons:

1. Considering the RWR feature matrix has a size of n x n, concatenating it to each node's feature vector will make the node features have a very high dimension (at least larger than n), which will introduce many parameters in the first layer of GCN. Besides, the time complexity and space complexity will also increase so that the model cannot easily be applied to real-world large-scale complex networks. Although Top-K sparsification of the RWR matrix is proposed in Section 5, the performance is sacrificed more or less. Besides, we should note that RWR itself has at least square time complexity, which requires lots of computation.

2. This paper argues that existing GCNs are only leveraging the graph structure in the form of the 2-hop or 3-hop neighborhood of
each node, and are not capable of extracting and exploiting long-range dependencies between nodes. However, there exist some works that can train GCN with deeper layers. For example, Cluster-GCN [1] can train GCNs with 5 layers, ie-HGCN [2] trains GCNs on HINs (heterogeneous information network) with 6-8 layers. Moreover, DeepGCNs [3] can successfully train very deep GCNs with as many as 112 layers.

3. This paper proposes to use k-step RWR to capture long-range structural information, and then concatenate the information to node features. The similar idea was adopted in existing work HAHE [4] as well, which also injects high-order (long-range) structural information as node features. What is more, the k-step RWR cannot be optimized by gradient descending, in an end-2-end fashion.

4. The length k of RWR is a crucial hyper-parameter, which plays a similar role to the number of layers of GCNs. However, I didn't find its setting in the experiments. Besides, it should be studied in the experiments to demonstrate its effectiveness of capturing long-range (larger than 2,3 hops) structural information.



Refs:

[1] [KDD 2019] [Cluster-GCN] An efficient algorithm for training deep and large graph convolutional networks
[2] [arXiv 2020] [ie-HGCN] Interpretable and Efficient Heterogeneous Graph Convolutional Network
[3] [ICCV 2019] [DeepGCNs] Making GCNs Go as Deep as CNNs
[4] [arXiv 2019] [HAHE] Hierarchical Attentive Heterogeneous Information Network Embedding
[5] [ICLR 2019] [GIN] How Powerful Are Graph Neural Networks

---

> ### Author Response · Authors · 2020-11-18
> **Answer to Reviewer 2**
>
> We thank the reviewer for his comments, and we answer to his concerns below.
> ### Q1
> The reviewer is correct, and we do present a discussion on the increase of parameters when adding RWR features in Section 5, but this addition of features does not happen when RWRReg alone is used. This is exactly the reason why we propose RWRReg as a practical method. In fact, RWRReg **does not** require any additional parameters. We would be happy to answer additional questions and provide additional clarifications on RWRReg if our comments do not make its description in the paper more clear. Furthermore, we add a new Appendix (Appendix D) in the paper showing an extremely fast way to calculate the RWRReg loss on a GPU. We also remark that the computation of RWR coefficients is an extremely well studied problem, and, as we write in our Related Work section, there have been proposed several efficient and fast methods for large scale graphs. There are even methods that can calculate the Top-k coefficients without having to compute them all (e.g. see [Wei et al., SIGMOD 2018])
>
> ### Q2
> In our paper we wrote 2-3 layers, as that is the common number of layers that is found in most GNN papers. However, these are not *strict* numbers, and oversmoothing is a general phenomenon that occurs when we increase the number of GNN layers, but the exact number depends on many things including the structure and size of the graph that the model is applied on.
> We provide below a brief discussion on the differences between our work and those mentioned by the reviewer.
> - Cluster-GCN is actually presenting a method for efficient training on large graphs (and it can also be paired easily with RWRReg as the latter it is just an additional term in the loss function during training). They consider very large graphs, and that is why they can reach 5-layers, but, as they show in Table 11, increasing the number of layers reduces performance.
> - ie-HGCN, as also remarked by the reviewer, works on heterogeneous graphs (which are something we do not consider), and even in their case, Figure 3b shows a  hard decrease in performance once the number of message-passing layers exceed a certain amount.
> - DeepGCN shows a method to train deep GCNs, but (1) it is not in the common semi-supervised scenario that is typically used for GNNs where the percentage of labelled nodes is less than 1% (which is the scenario we consider too for node classification), and (2) it works with point-cloud datasets, which have a structure which is very different from social and biological graph datasets like the ones commonly used for benchmarking GNNs.
>
> ### Q3
> The number of steps for the random walk is only used for the theoretical analysis. In practice, the RWR coefficients are obtained by reaching convergence (as RWRs are a Markov Process converging after a number of steps) and, hence, we can both theoretically and practically say we consider walks of infinite lengths. We do not understand the comment about gradient descent: RWRReg is fully differentiable as it only adds a regularization term during training. We thank the reviewer for pointing us towards HAHE, but we do not believe there are strong similarities. In fact, HAHE is focused on heterogeneous graphs, only works in the transductive setting (we consider the inductive too), and considers the case where important paths are given as input by the user (we do not assume this kind of information, and this is why we need a method like RWR to extract important relationships).
>
> ### Q4
> As written in the answer to the previous question, we kindly ask the reviewer to notice that RWRs are a process that reaches convergence, and in practice that is how the RWR coefficients are obtained (and what is commonly done, both in theory and in practice). We do not stop the walks at a certain length. With regards to the study of the effectiveness in capturing long range dependencies, Figure 1 in our paper shows that lower restart probabilities, which enable the exploration of nodes that are farther away, lead to higher results. Furthermore, the vast literature on RWR contains many works (which we reference in our paper; e.g. [Tong et al., ICDM 2006]) confirming this capability.

---

> > ### Comment · AnonReviewer2 · 2020-11-24
> > **About Q2 and Q3**
> >
> > Thanks for the reply. I am still not satisfied by response to Q2 and Q3.
> >
> > Q2:
> >
> > Very recently, a new method DAGNN towards deeper graph neural networks is proposed in [KDD 2020] which is able to adaptively incorporate information from large receptive fields (i.e. the long range structural information). The authors test their method DAGNN on datasets Cora, Pubmed and Citeseer, which are exactly the datasets used in this paper. The model depth experiment in Section 5.4 shows that DAGNN can explore very long range structural information (up to hundreds of hops). What is the difference of your method to this work?
> >
> > Q3:
> >
> > About "do not understand the comment about gradient descent": The walk length k needs to be specified by users, and the RWR score matrix S needs to be computed in advance. Therefore, the length k and structural proximities in S cannot be optimized during the training phase.

---

> > > ### Author Response · Authors · 2020-11-24
> > > **Clarifications on DAGNN and parameter K**
> > >
> > > We thank the reviewer for providing an answer to our rebuttal, and we further respond to his comments below.
> > >
> > > ### Q2
> > > Please note that, according to the ICLR reviewing guidelines (https://iclr.cc/Conferences/2021/ReviewerGuide), the paper proposing DAGNN should not be taken into account, since it was published after August 2 2020. This is a quote from the link above: “[..] if a paper was published on or after Aug 2, 2020, authors are not required to compare their own work to that paper.”
> > >
> > > We can however comment on the differences with our work. In short, DAGNN is very different form our contributions. First,  we *do not* aim at defining new state-of-the-art results, but at studying the amount of information that is lost by not considering long-range dependencies. Second, if we instead look at the methods, our method allows the model to take advantage of long-range dependencies without having to increase the number of layers, hence making it more practical. Third, note that in the DAGNN paper the authors only consider node classification, while we consider also graph classification and triangle counting. The addition of these two tasks allows us to provide more insights on the importance of long-range dependencies which are not available in the current literature.
> > >
> > >
> > > ### Q3
> > > We believe the reviewer might be confusing the random walk length with the parameter $k$ that we introduce for the Top-K selection in Section 5. These are two very different things, and we address both of them below.
> > >
> > > ##### Random Walk Length
> > > Please note that the length of the random walk with restart is *NOT* a parameter of our method. We calculate the RWR coefficients by reaching convergence, hence there is *no* notion of length of the walk. We provide also a link to the implementation on the popular library NetworkX, and a link to the original PageRank paper:
> > > - https://networkx.org/documentation/networkx-1.7/reference/generated/networkx.algorithms.link_analysis.pagerank_alg.pagerank.html
> > > - http://ilpubs.stanford.edu:8090/422/1/1999-66.pdf
> > >
> > > As can be seen, there is *no* parameter for the walk length. RWRs are a Markov process that reaches convergence after a certain number of iterations. By taking the results after convergence we are considering infinite length random walks and hence there is *no* notion of walk length.
> > >
> > > ##### Top-K Selection
> > > In Section 5 of our paper we show that if the graph we are considering is large (and we can't afford to keep all the RWR coefficients in memory), then we can take only the Top-K RWR coefficients for each node and still obtain an increase in performance (notice that in Figure 2, RWRReg with Top-K coefficients is always superior to GCN without RWRReg). Please notice that the parameter $k$ has nothing to do with the length of the random walks: RWR capture the importance of nodes, even ones that are far away (e.g. a node at distance 5 can have a higher RWR score than a node at distance 3). in fact we add a new appendix (Appendix E) confirming this property of RWR coefficients in a real-world scenario.
> > > In Section 5 we also show how different values of $k$ affect the results in Figure 2.
> > >
> > > We also highlight the work: "TopPPR: Top-k personalized pagerank queries with precision guarantees on large graphs.", Wei et al., SIGMOD 2018 that proposes a method that directly calculates the top-k coefficients (without having to calculate the whole matrix S) and that works efficiently for large-scale graphs.

---

### Official Review · AnonReviewer4 · 2020-10-28
**This work is quite constructive, but low innovation**

**Rating:** 6
**Confidence:** 4

**Review:**

Summarization

The authors formalize four levels of injection of graph structural information, and use them to analyze the importance of long-range dependencies. Among these four different structural information injections, the authors design various graph analysis tasks to evaluate the superiority of the proposed methods, and the experimental results could be reasonable and easy to follow


Strong points

1)This paper is good writing and easy to understood. The proposed Random Walk with Restart (RWR) Matrix and RWR Regularization are quite reasonable to boost the model performance of graph neural networks (GNNs), and easy to follow.

2)From the experimental results, the proposed methods have been proven efficient in various graph analysis tasks (node classification, graph classification and triangle counting) for different GNNs (GCN, GraphSage and GAT).


Weak points: The main weakness could be innovation and experiments

Innovation:

The proposed methods are quite heuristic, and I assume there could be many other improvements:

1)In section 2.2, the combination of Random Walk with Restart (RWR) Matrix and GCN is too heuristic, why not try other feature fusion methods rather than straightforward concatenation.

2)The developed RWR Regularization can be regarded as a formulation of Laplace Regularization, and only thing you do is replacing the Laplace matrix with the RWR matrix. Actually, there are also other graph construction methods (like consine similarity matrix etc) to replace the RWR matrix in RWR Regularization, you need to introduce additional experiments to prove the advantages of RWR matrix.

Experiments

1)The experimental results in Table. 1 are not so convinced. I agree with the point that your work don’t focus on defining new state-of-the-art results, but you still need to provide the node classification comparisons with the same train/validation/test split defined by Yang et al. (2016).

2)As your definitions, \lambda is a trade-off hyperparameter, but I miss the setting and ablation study of this important hyperparameter.

3)Why not try AD+RWRREG? From the results, this combination seems could be better (like GCN, Diffpool in Table.2).

Questions:

My questions have been included in Weak points part

Additional Feedback

1)Time complexity of constructing Random Walk with Restart (RWR) Matrix.

---

> ### Author Response · Authors · 2020-11-18
> **Answer to Reviewer 4**
>
> We thank the reviewer for his insightful comments, and we answer to his questions below.
> ### Q1
> We do not aim at new state-of-the-art results, we aim at studying the importance of long range dependencies. Reporting the results for only 1 train/val/test split would not give us much information on this regard, and this is why we use a different split at every iteration (and this of course leads to different results to what can be found in previous papers). Our results can further show the sensitivity of previous proposed methods to different splits of the data, and give more realistic results.
>
> ### Q2
> In our experiments we saw that minimizing RWRReg is much easier than minimizing the supervised loss for the downstream tasks (as we write in Section 4). We however notice that there is no need to perform particular tuning of such parameter, a value of 10^-6 has shown to work for all tasks and all models.
>
> ### Q3
> We would first like to iterate that the main contribution of our paper is the study of the importance of long-range dependencies. Our goal is not to obtain the new state-of-the-art, but to understand which are the limitation caused by not considering long-range dependencies. The reason why we take concatenation instead of more sophisticated methods, is that we want a clear and easy to interpret, method for studying the impact of long-range dependencies.
>
> ### Comment on similarity with Laplacian Regularization
> As explained in the previous answer, our aim is to study the impact of long-range dependencies. RWR have a long track record (see the references in our introduction section) of being able to extract nodes that are important to each other, even if they are far in the graph. The Laplacian matrix has no notion of distance between nodes, and hence would not provide us with any interesting information for the study of long range-dependencies.
>
> ### Comment on Time Complexity of RWR
> The construction of the RWR matrix is an extremely well studied problem, and we provide several references to fast implementations in the Related Work section. We also highlight the work: "TopPPR: Top-k personalized pagerank queries with precision guarantees on large graphs.", Wei et al., SIGMOD 2018 that proposes a method that directly calculates the top-k coefficients and that works for large-scale graphs.

---

### Official Review · AnonReviewer3 · 2020-10-28

**Rating:** 5
**Confidence:** 4

**Review:**

This paper studies GCNs when long-range dependencies have been added to the model as a regularizer. The regularizer proposed in this work is based on a random walk with restart (RWR) approach as RWR encourages the model to consider long-range dependencies. This paper shows that infusing the long-range dependencies using RWR regularizer improves the performance of some models for node classification and graph classification.

Here I list my concerns and questions about this work:

1. Baselines: As listed in the related work section, there are other works exploiting the long-range dependencies in the graph but this work only compares to the vanilla version of some models and shows adding the RWR regularizer improved their performance. However, it's already pointed out in the literature that long-range dependencies help the improvement. I expect this paper to compare with more strong baselines that also consider long-range dependencies so one can decide which approach better suits one application and gets better results.

2. Results in Table 1: Results reported for GCN and GAT are not consistent with the original papers. As an example, the GAT paper reports the accuracy of 72.5 ± 0.7% for node classification on the Citeseer dataset but this paper reports 66.4 ± 0.8 for GAT.

3. Results in Figure 1: The results reported in this table need more explanation. If smaller restart probability yields better accuracy, why not testing smaller than 5% for Cora in (a)? Why one of the figures has a start point of 5% on the x-axis and the other has 10%? I expected this figure to have low scores for low probabilities and then get better to some extent and then decrease after some point again. Can the authors elaborate more on why the performance is always decreasing as we increase the restart probability?

4. Section 5 on practical RWR regularization: This section studies a strategy for keeping the top-K RWR weights and sets the rest to 0 but the proposed strategy still needs to compute all RWR weights. Also, the results in Figure 2 show that larger values of K always yield better results. I was expecting that after some value of K the performance plateaus but the charts are always increasing and this shows that it's not a good idea to only keep the top-K weights.

---

> ### Author Response · Authors · 2020-11-18
> **Answer to Reviewer 3**
>
> We thank the reviewer for his insightful comments, and we answer to his questions below.
> ### Q1
> We would first like to iterate that the main contribution of our paper is the study of the importance of long-range dependencies. Our goal is not to obtain the new state-of-the-art, but to understand which are the limitation caused by not considering long-range dependencies. Furthermore, most previous methods are limited to transductive settings (e.g. Geom-GCN), while we perform also inductive tasks, which would make the comparison impossible.
>
> ### Q2
> As mentioned in the previous answer, we do not aim at new state-of-the-art results, we aim at studying the importance of long range dependencies. Reporting the results for only 1 train/val/test split would not give us much information on this regard, and this is why we use a different split at every iteration (and this of course leads to different results to what can be found in previous papers). Our results can further show the sensitivity of previous proposed methods to different splits of the data, and give more realistic results.
>
> ### Q3
> Smaller restart probabilities means that the RWR can explore more distant nodes, and hence capture more information that is not available to GNNs (and this explains the better results). The reason why we start at different percentages is simply because the iterative method we use to calculate the RWR coefficient requires a minimum restart probability to reach convergence that depends on the considered graph.
>
> ### Q4
> It is actually not necessary to calculate all the RWR coefficients as the reviewer says. The computation of the RWR coefficients is an extremely well studied problem with many efficient and fast methods. A very efficient method (which we provide in the Related Work section) to only calculate the top-k coefficients can be found in: "TopPPR: Top-k personalized pagerank queries with precision guarantees on large graphs.", Wei et al., SIGMOD 2018.
> It is true that keeping all the RWR coefficients leads to higher results, however, when it is not possible for memory reasons, we show that even a subset of them leads to higher performance (notice that in Figure 2, RWRReg with top-k selection is *always* superior to GCN without RWRReg).

---

### Official Review · AnonReviewer1 · 2020-11-09
**Incremental contribution**

**Rating:** 4
**Confidence:** 4

**Review:**

The paper considers the long-range dependency and proposes four levels of injection of longer-range graph structure information based on random walks with restart (RWR). Experimental results show that the proposed models perform well on the tasks of node classification, graph classification, and counting triangles.
Utilizing long-range dependency is not new in graph neural networks. I do not think the authors give enough reviews about important related tasks; their related work section focuses more on RWR. Considering the motivation and the solved issues of graph neural networks, more relevant literature in GNN domains should be added, such as MixHop (Abu-El-Haija et al., 2019), Snowball (Luan et al., 2019), APPNP (Klicpera et al., 2019 ), GDC (Klicpera et al., 2019). Compared with those works, the RWR regularization seems incremental. Adding the RWR features is just a new feature and adding the RWR regularization term actually can be translated into a kind of message passing schema.
Moreover, some details of the method are not very clear. For example, how do we calculate $S_{I,j}$? When we add the regularization, do we use all node pairs or just the node pairs within some distance? If using all node pairs, is the computational complexity too high?

---

> ### Author Response · Authors · 2020-11-18
> **Answer to Reviewer 1**
>
> We thank the reviewer for the insightful comments and we answer to his questions below.
> ### Q1
> We would first like to iterate that the main contribution of our paper is the study of the importance of long-range dependencies. Our goal is not to obtain the new state-of-the-art, but to understand which are the limitation caused by not considering long-range dependencies. The methods shared by the reviewer present some similarities to our work, but have substantial differences that we showcase below.
> - MixHop uses different powers of the adjacency matrix to allow communications between nodes at multiple distances. This has several limitations: (1) the maximum distance considered is still limited to the maximum considered power, while our method can explore distances of any length. (2) To consider nodes at distance k, MixHop requires to store in memory k adjacency matrices (or to calculate them on the fly), while our method only requires the original adjacency matrix, and the RWR matrix (which are of the same size). (3) MixHop still incurs in over-smoothing when multiple layers are applied, while our method works with few GNN layers, hence avoiding over-smoothing.
> - Snowball stacks the representation of a node at each layer to generate the final embeddings. This method does not avoid oversmoothing in any way, as the representations at every "higher" layer will suffer from it, just like a regular GCN.
> - APPNP and GDC both use a diffusion strategy based on RWR instead of aggregating information from neighbours. This is like creating smart or extended neighbourhoods which are then used for node aggregation, while we in our work we show that node aggregation (or message-passing) without additional information (e.g., RWR features or RWR-based regularization) is not capable of fully exploiting structural graph information.
>
> ### Q2
> S_{i,j} is the probability that a random walk with restart starting at node i, ends in node j. This quantity is hence obtained with the computation of the RWR coefficients. As written in our Related Work section, this is a very well studied problem with a large number of very fast methods to perform this task.
>
> ### Q3
> In our experiments shown in Table 1, Table 2, Table 3, Figure 1, we use all node pairs. In Section 5 we *do not* use all pairs, as we only keep the top-k pairings for each node. We also agree with the reviewer that this can be an interesting direction for future work.
> We talk about computational complexity in Section 5, and in fact, as suggested by the reviewer, in Section 5 we do not use all node pairs. Nevertheless, while it might seem very expensive, it is also possible to obtain the value of the RWRReg objective with simple matrix multiplications by taking advantage of the properties of the graph Laplacian matrix. This method is extremely fast on GPUs. We added a section on this in the new version of the paper in Appendix D.

---

### Author Response · Authors · 2020-11-18
**Comments & Updates for Reviewers**

We thank the reviewers for their insightful comments. We did our best to provide answers to all the concerns, and we are fully available to provide further comments.

There are a few general comments we would like to remark, as we believe they can be helpful for the reviewers while reading our rebuttal and re-reading the paper:
- Our main contribution is the study of the importance of long-range dependencies. As written in the Introduction, we do not aim at defining new state-of-the-art results. We believe a paper that studies the weaknesses and limitations of existing models can be as useful to the community as a paper that defines a new state-of-the-art.
- RWRReg is a regularization term that makes use of the random walk with restart coefficients. The models trained with RWRReg do not require any additional parameter, or any additional operation at inference time. In fact, the inference time and the number of parameters for a model trained with RWRReg are *exactly* the same as the inference time and the number of parameters of a model trained without it.
- Random walks with restart are an extremely well studied problem. In the Related Work section we reference many existing works proposing extremely efficient methods to compute random walk with restart coefficients, even for massive graphs. There further are works that can compute directly the top-k coefficients without the need to compute them all.

Finally, we have update the paper with an appendix (Appendix D) showing how it's possible to take advantage of the great velocity of GPUs in computing matrix multiplications to efficiently compute the RWRReg loss term.

---

> ### Author Response · Authors · 2020-11-24
> **Further Updates**
>
> We further added a new revised version of the paper with an additional appendix (Appendix E) showing an empirical analysis of the ability of random walks with restart to capture long range dependencies.

---

### Decision · Program_Chairs · 2021-01-07
**Final Decision**

**Decision:**

Reject

**Comment:**

R4 of this submission was slightly positive on this submission while all other reviewers expressed quite significant concerns in their reviews. R4 also agreed that the originality and experimental results as presented in this submission are not sufficient during discussion, although he/she pointed out the incorporation of long-range structural information is novel. Given the above recommendations and discussions, a reject is recommended.